# Multiple Sclerosis and Autoimmune Comorbidities

**DOI:** 10.3390/jpm12111828

**Published:** 2022-11-03

**Authors:** Viviana Nociti, Marina Romozzi

**Affiliations:** 1Centro Sclerosi Multipla, Fondazione Policlinico Universitario ‘Agostino Gemelli’ IRCCS, 00168 Rome, Italy; 2Università Cattolica del Sacro Cuore, 00168 Rome, Italy

**Keywords:** multiple sclerosis, personalized medicine, autoimmune, comorbidities

## Abstract

Multiple sclerosis (MS) is a chronic inflammatory and neurodegenerative disease of the central nervous system characterized by broad inter- and intraindividual heterogeneity and different prognoses. Multisystem comorbidities are frequent features in people with MS (PwMS) and can affect treatment choices, quality of life, disability and mortality. In this scenario, autoimmune comorbidities play a cardinal role for several reasons, such as the implication on MS pathogenesis, diagnostic delay, disease activity, disability progression, brain atrophy, and treatment choice. However, the impact of an autoimmune comorbid condition on MS is not fully elucidated. This review aims to summarize the currently available data on the incidence and prevalence of autoimmune diseases in PwMS, the possible effect of this association on clinical and neuroradiological MS course and its impact on treatment choice.

## 1. Introduction

Multiple sclerosis (MS) is a chronic, inflammatory and degenerative demyelinating disease of the central nervous system (CNS), of unknown etiology, affecting individuals in early adulthood [1]. The disease is characterized by broad inter- and intraindividual heterogeneity with various clinical presentations, different subtypes and prognoses and consequently variable responses to treatments [1]. MS treatment guidelines have changed significantly over the past decades due to the growing number of approved disease-modifying therapies (DMTs), which can influence the disease course by preventing relapses and disability progression in people with MS (PwMS) [2].

Although the etiology and pathogenesis of MS remain unclear, it is believed to be an immune-mediated disease. The pathophysiology of MS seems to be characterized by an aberrant immune activation [3]. This immune dysregulation leads to neuroinflammation in which both immune cells from the periphery and resident cells of the CNS (e.g., microglia and astrocytes) are involved [3]. 

MS may share similar underlying pathogenesis with certain comorbidities, particularly autoimmune pathologies, which may arise from genetic susceptibility to autoimmunity and overlapping pathogenetic mechanisms [4].

Comorbidity is defined as the co-existence of multiple diseases or medical conditions in a patient, which may share similar overlapping pathophysiology and affect the course of the disease itself [5]. 

Multisystem comorbidities are frequent features in PwMS, particularly neurological disturbance, other comorbidity autoimmune conditions, psychiatric comorbidities, cardiovascular diseases, chronic lung diseases and metabolic disorders. Among neurological disturbances, epilepsy, migraine and restless leg syndrome are more frequent in patients with MS compared with the general population [6].

Psychiatric comorbidities (i.e., depression, anxiety, and bipolar disorder) have been associated with poor quality of life (QoL), reduced adherence to DMTs and fatigue [7].

Cardiovascular comorbidities such as abnormalities in blood pressure, heart rate, heart rhythm, and left ventricular systolic function are common in PwMS, and a diagnosis of MS increases the risk of myocardial infarction, stroke and heart failure [8,9]. 

Many comorbidities are present before or at MS symptom onset, but it seems that the prevalence increases over time [10]. For example, some comorbidities such as depression and anxiety may represent preclinical symptoms of MS resulting from immunological and inflammatory changes in the CNS [7]. 

Over the last decades, the average age of the MS population has increased correlating with increasing general life expectancy, advances in DMTs, the adoption and subsequent widespread use of magnetic resonance imaging (MRI) and improved health and social care for these patients. Further, MS has become a lifelong condition. The risk of multi-morbidity increases with age and in patients with a chronic disease such as MS [11]. However, over time, the accrual of comorbidities can be explained by several other factors such as physical invalidity, a background of common risk factors and the use of DMTs [4].

Several studies have shown that the presence of comorbidities can delay diagnosis. Furthermore, comorbidities can affect treatment choices, QoL, disability and mortality [12]. 

Marrie et al., using the North American Research Committee on Multiple Sclerosis Registry, evaluated the association between comorbidities and both the diagnostic delay and severity of the disability at MS diagnosis. They found that the presence of comorbidities was associated with greater diagnostic delays and increased disability at diagnosis [13]. The presence of comorbid conditions increases the complexity of patient management and bears important clinical and socioeconomic implications for PwMS [12].

This review aims to describe the current evidence regarding the range of comorbidities in PwMS, focusing on autoimmune comorbidities, and widen the current knowledge about the influence of these comorbidities on the clinical features and therapeutic choices in MS.

## 2. Neurological Comorbidities and MS

Several studies have suggested that comorbid neurological conditions such as epilepsy and migraine are more frequent in PwMS [6].

### 2.1. Epilepsy

Epileptic seizures are more common in PwMS than in the general population. A systematic review found that the incidence of seizure disorders (18 studies) ranged from 0.6% to 6% after MS onset, and the prevalence (24 studies) ranged from 0.9% to 8%, both higher than reported in the general population. However, heterogeneity among these studies was substantial [14].

Seizures may occur at any point during the course of MS and have also been described as the presenting symptom of MS [15,16,17]. Seizures have been observed in patients with relapsing-remitting MS (RRMS) as well as in progressive forms [15,18]. 

Several studies suggested that focal onset seizures are more common than generalized seizures and secondary generalization of focal seizures is also frequent [19,20].

A systematic review demonstrated that patients with MS who had seizures are more likely to have a younger age of onset compared to MS patients without seizures, and there was also a trend demonstrating higher disability scores in patients with epilepsy [18]. A longitudinal study by Calabrese et al. showed an association between the severity of cortical pathology, namely the number and volume of cortical lesions evaluated by double-inversion recovery (DIR) imaging, and the occurrence of epilepsy in PwMS. Moreover, in patients with RRMS and epilepsy, cognitive dysfunction was four-times higher than in the RRMS group without epilepsy [21]. 

In a study by Martinez-Lapiscina et al., MS patients with seizures had a higher mean number of cortico-juxtacortical lesions on T2-weighted/fluid attenuation inversion recovery magnetic resonance imaging as compared to MS controls [22]. 

The pathologic substrate of epilepsy in MS is not fully elucidated. Still, the involvement of grey matter, along with brain inflammation, are presumably the reasons behind the increased incidence of epilepsy [23].

### 2.2. Migraine

Migraine affects up to 50% of PwMS with a significantly higher prevalence than in the general population [24].

PwMS are more than twice as likely to report migraine than controls, with a significant association for migraine without aura [25]. Migraine can precede MS onset, be associated with relapses, or manifest during the disease course [26,27]. However, the prevalence of comorbid migraine seems to be higher in patients with RRMS compared to those with progressive forms [28].

The mechanism explaining the association between MS and migraine remains to be clarified, but several hypotheses have been postulated. The inflammatory demyelinating MS lesions may induce migraine by disrupting the pathways involved in the pathogenesis of migraine, deputed to pain stimuli processing. It was observed that lesions within the midbrain, especially in the periaqueductal grey matter, were frequently associated with comorbid migraine in PwMS [29,30].

Furthermore, a study on rodent models of autoimmune-induced cortical demyelination showed that cortical demyelination was associated with accelerated cortical spreading depression (CSD), which has been implicated in migraine pathophysiology, both with and without aura [31,32]. 

Only a few pieces of evidence support the hypothesis that CSD may lead to a subtle increase in permeability of the blood–brain-barrier and neuroinflammation, thereby exposing antigens in the CNS to circulating T cells [33]. Graziano et al. found an increased frequency of contrast-enhancing lesions in MS patients with migraine, specifically within the RRMS disease subtype, and postulated that having migraine comorbidity may increase the level of blood–brain-barrier disruption in these patients [27]. 

Regarding the overall impact on MS severity, some studies have found no significant correlation between the level of disability and the presence of migraine [28], while the presence of comorbid migraine negatively affects some aspects of QoL in these patients [34]. 

## 3. Autoimmune Comorbidities and MS

The coexistence of autoimmune/inflammatory comorbidities has recently engendered significant interest in the MS research field. First, MS and specific comorbidities may have underlying pathogenesis. Second, the association between autoimmune comorbidities and MS may reveal common genetic and environmental risk factors [4].

For example, smokers with multiple sclerosis had an increased risk of developing comorbid autoimmune disease after MS onset [35].

Several articles describe the association between autoimmune/inflammatory conditions and MS, but the results are still inconsistent. A population-based case-control study on autoimmune comorbidities and MS showed that before MS diagnosis, uveitis occurred 3-fold more often, and inflammatory bowel disease (IBD) occurred 1.7-fold more often in patients with MS [36]. A nationwide cohort study in Denmark found that PwMS were at an increased risk of developing ulcerative colitis and pemphigoid [37]. Nevertheless, other studies, including a recent multicentric population-based study, did not find any increased risk of autoimmune diseases among MS patients [38].

A systematic review of the incidence and prevalence of autoimmune disease in PwMS found that psoriasis and thyroid disease were the most prevalent autoimmune comorbidities. The findings also supported an increased risk of IBD, uveitis, and pemphigoid [39]. However, the authors stated that they could not draw conclusions because the study was mainly heterogeneous with respect to the populations studied, methods of ascertaining comorbidities, and reporting of findings [39]. 

A systematic review and meta-analysis on the overall risk of other autoimmune diseases in PwMS and their first-degree relatives showed that the OR of thyroid disease was increased in both PwMS (OR 1.66) and their relatives (OR 2.38). A comparable association was observed between MS and IBD (OR 1.56) and psoriasis (OR 1.31; *p* < 0.0001), though not in relatives [40]. 

Table 1 lists relevant studies on MS and concomitant autoimmune conditions.

### 3.1. Type 1 Diabetes Mellitus

The co-occurrence of type 1 diabetes mellitus (DM) and MS is supported by several studies. A population-based cohort study found that patients with type 1 DM were at 3-fold greater risk for developing MS. Furthermore, the risk for type 1 DM in first-degree relatives of patients with MS was increased by about 40% [41]. The authors suggested that similarities in immunological features between type 1 DM and MS and/or unknown environmental factors might contribute significantly to the co-occurrence of these two diseases [41]. Similarly, a Sardinian study observed a 5-fold and 2-fold higher prevalence of type 1 DM in patients with MS and their first-degree relatives compared with the general population. It concluded that common genes might contribute to susceptibility to both diseases [42].

Furthermore, Bechtold et al. conducted a cohort study on a pediatric and adolescent population affected by type 1 DM in Germany and Austria. They demonstrated a considerably higher risk of MS co-occurrence in the diabetic population [43].

### 3.2. Autoimmune Thyroid Disease

Autoimmune thyroid disease seems to affect PwMS and their first-degree relatives more than the general population, regardless of treatment with IFN-β and alemtuzumab, but the results are variable. A controlled prospective study found that thyroid disorders were at least three-times more common in women with MS than in female controls [44]. Another study on 353 patients not treated with IFN-β found a statistically significant higher prevalence of autoimmune thyroiditis in male MS patients compared with male controls but not in female patients [45]. In a study on a Spanish cohort of 93 untreated PwMS, MS patients had a higher prevalence of antithyroid antibodies compared with the general population [46].

### 3.3. Inflammatory Bowel Disease

Previous studies have established an association between MS and IBD, including Crohn’s disease and ulcerative colitis (UC) [47,48]. A systematic review and meta-analysis showed that IBD and PwMS seem to be associated with a 50% increased risk of MS or IBD comorbidity, respectively, with no apparent differences between patients with Crohn’s disease or ulcerative colitis [49]. 

### 3.4. Psoriasis

An association between MS and psoriasis has not been clarified, and several studies have tried to determine whether an association exists. Marrie et al. found that the risk of incident psoriasis was 54% higher in PwMS (HR 1.54; 95%CI: 1.07–2.24) [50]. One case-control study investigating whether patients with a diagnosis of MS had higher rates of concomitant psoriasis found a higher-than-expected frequency of psoriasis among PwMS [51]. In contrast, other studies did not corroborate the association between psoriasis and MS [52]. In a study on 658 consecutive patients, the prevalence of psoriasis in MS patients compared to the general population did not differ significantly [48]. Similarly, a large multicenter study on autoimmune disease risk in MS patients found that the frequency of psoriasis in MS patients did not differ from spousal controls (5.8% of the MS population and 5.4% of controls) [38]. The association between the two conditions may reflect shared genetic, environmental factors and immune pathways, and the effectiveness of fumarates in both conditions may yield etiologic insights into MS [53].

**Table 1 jpm-12-01828-t001:** Relevant studies on the co-occurrence of MS and autoimmune diseases.

Author (Date) [Reference]	Concomitant AID	Study Population	Main Findings
Nielsen (2006) [41]	Type 1 DM	11,862 PwMS, 6078 type 1 DM cases	RR for MS in type 1 DM patients: 3.26 (95% CI: 41.30–7.97)
Marrosu (2002) [42]	Type 1 DM	1090 PwMS, 2180 parents, 3300 siblings	Type 1 DM prevalence in MS is 3-fold greater than in healthy siblings and 5-fold the general population
Bechtold (2014) [43]	Type 1 DM	19 PwMS, 56,653 type 1 DM cases	RR of MS in type 1 DM patients: 3.35 (95% CI: 1.56–7.21) to 4.79 (2.01–11.39)
Edwards (2005) [48]	Type 1 DM	658 PwMS	Type 1 DM prevalence of 0.9%; OR 18.14 (95% CI: 6.40–2.17)
Ramagopalan (2007) [38]	UC	5032 PwMS, 2707 controls	No difference between PwMS and controls
Roshanisefat (2012) [47]	UC	20,276 PwMS, 203,951 controls	Increased risk in PwMS: HR 1.49 (95% CI: 1.22–1.82)
Roshanisefat (2012) [47]	CD	20,276 PwMS, 203,951 controls	Increased risk in PwMS: HR 1.45 (95% CI: 1.17–1.81)
Edwards (2005) [48]	IBD	658 PwMS	IBD prevalence of 1.2%; OR 3.17 (95% CI: 6.40–2.17)
Farez (2014) [52]	CD	211 PwMS, 211 controls	No difference between PwMS and controls
Ramagopalan (2007) [38]	Psoriasis	5032 PwMS, 2707 controls	No difference between PwMS and controls
Roshanisefat (2012) [47]	Psoriasis	20,276 PwMS, 203,951 controls	Increased risk in PwMS: HR 1.73 (95% CI: 1.42–2.10)
Edwards (2005) [48]	Psoriasis	658 PwMS	No higher prevalence compared to controls
Marrie (2017) [50]	Psoriasis	4911 PwMS, 23,274 controls	Risk of psoriasis was 54% higher in PwMS (HR 1.54; 95%CI: 1.07–2.24).
Fellner (2014) [51]	Psoriasis	214 PwMS, 192 controls	Psoriasis prevalence of 4.21%; OR: 8.39 (95% CI: 1.05–66.81)
Farez (2014) [52]	Psoriasis	211 PwMS, 211 controls	No difference between PwMS and controls
Niederwieser (2003) [45]	AITD	353 PwMS, 308 controls	Higher prevalence of AITD in male MS patients (9.4%) than in male controls (1.9%; *p* = 0.03)
Edwards (2005) [48]	AITD	658 PwMS	AITD prevalence of 3.2%; OR 1.80 (95% CI: 3.02–1.07)

AID, autoimmune disease; AITD, autoimmune thyroid disease; CD, Crohn’s disease; CI, confidence interval; DM, diabetes mellitus; HR, hazard ratio; IBD, inflammatory bowel disease; OR, odds ratio; PwMS, people with multiple sclerosis; RR, relative risk; UC, ulcerative colitis.

### 3.5. Drug-Related Autoimmune Disorders

The possibility of comorbid autoimmune disorders developing secondary to DMTs has also been a concern. Autoimmunity following alemtuzumab therapy is a well-recognized adverse effect of alemtuzumab, with thyroid disease being the most common in more than a third of patients. Other antibody-mediated autoimmune disorders are associated with alemtuzumab, including idiopathic thrombocytopenic purpura and anti-glomerular basement membrane disease [54]. IFN-β might be associated with an increased risk of thyroid autoimmunity and dysfunction, particularly within the first year of treatment [55].

## 4. Effects of Autoimmune Comorbidities on the Clinical and Neuroradiological Course of MS

Several studies investigated the effect of different autoimmune conditions on the clinical course of MS. Patients with MS, and a concurrent autoimmune disorder seem to have lesser disabilities compared to those with isolated MS after an average of 5.91 years. It has been suggested that the concurrent autoimmune disorder may increase body tolerance against autoantigens [56]. A study performed by Fanouriakis et al. on 9 patients with both systemic lupus erythematosus and MS showed that the coexistence of the two diseases does not seem to be associated with a severe phenotype for either entity. In particular, systemic lupus erythematosus remained quiescent in all patients while on standard immunomodulatory MS therapy [57]. A study on 66 patients with concomitant MS and IBD showed that these patients have a milder course of the disease than patients with MS alone, after a median of 12 years of disease duration [58].

Regarding the effects of comorbid autoimmune diseases on the radiological outcome, Zivadinov et al. analyzed magnetic resonance (MR) imaging findings in MS patients with autoimmune comorbidities. They found that comorbidities in patients with MS were associated with more severe MR imaging outcomes of demyelination and neurodegeneration, evidenced by several nonconventional MR imaging measures, including brain atrophy, magnetization transfer imaging, and diffusivity. The findings were significant for psoriasis, type 2 DM, and thyroid disease [59]. However, the study included 40 patients with type 2 DM, which is considered more a metabolic-acquired disease rather than an autoimmune condition, and only 3 patients with type 1 DM.

Lorefice et al. performed a case-control MRI study on 286 PwMS, of which 30 subjects had DM type 1, 53 patients had autoimmune thyroiditis, and 4 had celiac disease. Multiple regression analysis found an association between type 1 DM and lower grey matter and cortical grey matter volumes, independent from MS clinical features and related to type 1 DM duration [60].

The correlation between brain atrophy and autoimmune comorbidities may have significance in the progression of disability, as several reports have shown that brain and spinal atrophy may represent reliable biomarkers of neurodegeneration that correlate with physical and cognitive impairment in PwMS [61,62].

## 5. Effect of Autoimmune Comorbidities on Treatment Choices 

The treatment choice in PwMS and concomitant autoimmune comorbidities that share common immunological interfaces is still challenging. Numerous therapeutic strategies are either approved for different autoimmune disorders or may be repurposed for several diseases, with the possibility of using a drug that can be effective in both conditions. Contrarily, other medications may exacerbate pre-existing autoimmune disorders or trigger their onset. For example, tumor necrosis factor-alpha (TNFα) blockers, established as effective agents in the treatment of several autoimmune conditions such as rheumatoid arthritis, psoriasis and IBD, can induce or worsen demyelinating diseases and are contraindicated in the treatment of MS [63]. 

Similarly, interferons can unmask silent autoimmune processes or induce de novo autoimmune diseases, and their use in MS with concomitant autoimmune diseases is not recommended [64,65,66].

Alemtuzumab is not recommended in the case of MS and autoimmune comorbidities for its association with autoimmunity. The cause of autoimmunity is not entirely understood and is likely related to the pattern of T- and B-cell repopulation after their depletion [67].

Natalizumab, a monoclonal anti-α4 antibody, inhibits the trafficking of lymphocytes from the blood into CNS. There have been several cases of patients with MS treated with natalizumab who developed rheumatoid arthritis or experienced an onset or exacerbation of psoriasis [68,69]. Natalizumab modifies the composition of lymphocyte subpopulations and alters the migration of leukocytes across the blood–brain-barrier, shifting the inflammatory response from the CNS towards other tissues [70]. Furthermore, natalizumab has been demonstrated to be an effective drug for the induction and maintenance of remission in patients with Crohn’s disease [71,72]. Therefore, natalizumab may be an alternative treatment option for patients with MS and Crohn’s disease but should be used cautiously in patients with comorbid psoriasis or rheumatoid arthritis.

Anti-CD20 antibodies used for MS include rituximab, ocrelizumab and ofatumumab. Ocrelizumab has been demonstrated to be an effective and safe treatment option for rheumatoid arthritis [73]. In contrast, ocrelizumab and rituximab were associated with an increased risk of developing psoriasis and IBD. Therefore, it should be avoided in patients with MS plus psoriasis or IBD [74,75,76].

The sphingosine-1-phosphate receptor modulators (fingolimod, siponimod, ozanimod) are potentially useful in IBD, rheumatoid arthritis and psoriasis, as demonstrated in animal models and few reports [77,78,79]; however, further applications in clinical trials are still needed to ascertain the effectiveness in these pathologies. Ozanimod has been proven safe and effective in patients with moderate to severe ulcerative colitis and may represent an option for patients with concomitant highly active MS and ulcerative colitis [80].

Among first-line therapies, dimethyl fumarate enhances the nuclear factor erythroid 2 related factor 2 (Nrf2) transcriptional pathway. It is approved for both relapsing-remitting mild/moderate MS and moderate/severe psoriasis and represents the treatment of choice in case of co-occurrence of these pathologies [81,82].

The DMTs approved for multiple sclerosis and other common autoimmune conditions are synthesized in Table 2. A pragmatic treatment approach to MS and autoimmune disorders should foresee first a check for contraindications, second, an evaluation of the disease/severity of the concomitant conditions, to tailor therapies for the individual patient [83].

## 6. MS and Autoimmune Comorbidities: Unmet Needs

The association between MS and different comorbidities is of rising interest as a factor that could explain the heterogeneity of outcomes. 

Over the past decades, insufficient attention has been paid to comorbid conditions in MS, as it was considered mainly a disease of young adults with a limited comorbidity burden. Recent studies demonstrated the importance of investigating comorbidities in PwMS, as several comorbid conditions influence the course of the disease, disability progression, worsened QoL, treatment choices and mortality [84,85,86]. Another important point is the diagnostic delay in patients with comorbidities [13]. An explanation could be that the clinician could erroneously attribute MS symptoms to a preexisting condition, increasing the time from symptom onset to diagnosis and subsequently disability at diagnosis [13].

However, studies investigating comorbidities in PwMS have been performed mainly in small cohorts, and a global view of comorbidity in MS is lacking. A clear understanding of the risk of developing these pathologies and their prevalence in PwMS is necessary [6].

Furthermore, patients with comorbidities are usually excluded in clinical trials of disease-modifying therapies in MS. Thus, trial results may not be representative of a real-world scenario and cannot be applied to a standard clinic population with general comorbidities [87]. However, comorbidities may influence disease activity, disability progression and treatment choices [12]. 

Marrie et al. reviewed the published results of nine placebo-controlled trials of DMTs for MS. Of the nine trials evaluated, five excluded individuals with different comorbidities, and the explanation of the exclusions for comorbidities was unclear in four trials. None of the trials reported the comorbidity status of patients at enrollment [87]. 

To address this knowledge gap, the International Workshop on Comorbidity in Multiple Sclerosis produced several recommendations for future clinical trials, indicating relaxing restrictions on the inclusion of patients with comorbidity. The comorbidities prioritized were depression, anxiety, autoimmune disease, diabetes, cancer, hypertension, and migraine [87].

In this scenario, autoimmune comorbidities play a cardinal role for several reasons. First, the coexistence of autoimmune disorders may reflect common underlying pathobiology. Second, as the treatment landscape continues to expand with the development of new DMTs, concomitant pathologies may present a unique therapeutic target. 

For example, MS and type 1 DM share T-cell-mediated autoimmunity [88]. Type 1 DM is caused by T lymphocytes’ destruction of the insulin-producing β-cells in pancreatic islets once activated by particular insulin epitopes on antigen-presenting cells. In vitro, T-cells from type 1 DM patients reacted to pancreatic islet and CNS antigens [89]. Children with concomitant CNS demyelination and type 1 DM exhibited heightened T-cell reactivities to self-antigens, and these responses were not strictly limited to the disease target organs [90].

The presence of shared pathways and the involvement of similar cell types in the pathogenesis of autoimmune conditions may underlie the risk of both MS and other autoimmune diseases [91]. Furthermore, genetic loci shared between multiple autoimmune diseases are involved in a wide range of immune pathways (i.e., T-cell activation, B-cell activation, cytokine signaling), helping to explain common pathogenic features [92]. Apart from loci within the major histocompatibility complex (MHC) associated with the greater risk, multiple non-MHC genes mainly involved in the regulation of immune response may also explain the susceptibility of multiple autoimmune conditions [92].

However, the impact of an autoimmune comorbid condition on MS remains yet to be fully elucidated. The coexistence of another autoimmune disease may reduce the possibility of disability progression due to unknown changes in immunological pathways, increasing the tolerance against autoantigens. Alternatively, regulatory T-cells can prevent the inflammatory activity of Th1 and Th17 cells following the secretion of anti-inflammatory cytokines. The activation of more regulatory T-cells and the production of more anti-inflammatory cytokines could be another hypothesis for a milder course in patients with two concomitant autoimmune diseases. Further studies with larger sample sizes are required to confirm the finding of brain atrophy associated with autoimmune comorbidities.

## 7. Conclusions

In this comprehensive review, we summarized currently available data on the incidence and prevalence of autoimmune diseases comorbidity in PwMS, the possible effect of this association on clinical and neuroradiological MS course and its impact on treatment choice. We also highlighted the unmet needs in this field. The currently available data make it difficult to draw firm conclusions concerning the extent of coexistence of other autoimmune diseases in PwMS and all other topics discussed herein. Further studies in this area are warranted considering the implication on MS pathogenesis, the possible role of autoimmune co-morbidity on MS diagnosis delay, disease activity and disability progression, brain atrophy, and treatment choice.

## Figures and Tables

**Table 2 jpm-12-01828-t002:** DMTs approved for MS and their use in other autoimmune disorders.

DMTs	MS	IBD	PsO	RA
INF	+	−	−	−
DMF	+	−	*	−
TER	+	−	−	*
S1PR Ozanimod	+	* + (UC)	−	−
CLAD	+	−	−	−
NAT	+	+	−	−
ALE	+	−	−	−
Anti-CD20	+	−	−	+

+, FDA approved; * used off-label; −, not used; ALE, alemtuzumab; Anti-CD20: CD20 antibodies; AZA, azathioprine; CLAD: cladribine; DMF, dimethyl fumarate; FDA, US Food and Drug Administration; IBD, inflammatory bowel disease; IFN, interferon; MS, multiple sclerosis; MTX, methotrexate; NAT, natalizumab; PsO, psoriasis; RA, rheumatoid arthritis; S1PR, sphingosin-1-phosphate receptor modulator; SLE, systemic lupus erythematosus; TER teriflunomide; UC: ulcerative colitis.

## Data Availability

Not applicable.

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
