# Peer review of "Multiple Sclerosis and Autoimmune Comorbidities"

_jpm, 2022, doi:10.3390/jpm12111828_

Round 1

Reviewer 1 Report

Nice paper regarding comorbidities but I have 2 remarks: 

1. for a review, the paper is relatively short, additional tables listing studies and the number of patients would be appropriate. 

2. neurological comorbidities are lacking, authors should add this paragraph as well 

Author Response

Nice paper regarding comorbidities but I have 2 remarks:

  1. for a review, the paper is relatively short, additional tables listing studies and the number of patients would be appropriate.

We thank the Referee for her/his important suggestion, and accordingly, we added a table (Table 1) listing the most relevant studies of concomitant MS and autoimmune conditions. All the changes are marked in red font.

  1. neurological comorbidities are lacking, authors should add this paragraph as well

We thank the Referee for raising this important point, and accordingly, we have added a paragraph focused on MS and neurological comorbidities entitled “Neurological comorbidities and MS”. All the changes are marked in red font.

Reviewer 2 Report

This narrative review is concise and provides general information on autoimmune comorbidities of multiple sclerosis. However, organization and presentation of data is misleading.

Suggest to put headings (e.g. endocrine, skin, systemic, drug-related, etc.) in the second section (Autoimmune comorbidities and MS) because e.g. psoriasis is mentioned in three different paragraphs of this section. Presented data should be organised condition-wise in the same paragraph but not chaotically.

Author Response

Reviewer #2 reports:

This narrative review is concise and provides general information on autoimmune comorbidities of multiple sclerosis. However, organization and presentation of data is misleading.

Suggest to put headings (e.g. endocrine, skin, systemic, drug-related, etc.) in the second section (Autoimmune comorbidities and MS) because e.g. psoriasis is mentioned in three different paragraphs of this section. Presented data should be organised condition-wise in the same paragraph but not chaotically.

We thank the Referee for her/his important suggestion. Accordingly, we added headings in the section autoimmune comorbidities and MS, and we reorganized the order of the presented data. We also added a separate paragraph on psoriasis. The changes are marked with red font.

Reviewer 3 Report

This review discusses the association of MS with other autoimmune disease co-morbidities. Given the limited and conflicting literature in this field it is a challenging undertaking to write a cohesive review on this topic. That being said, there could be substantial improvement in the organization and structure of the review. There is a lot of repetition that could be minimized to make the review much shorter and concise (making it an "easier read"). 

There are many stand alone thoughts or paragraphs. Could this be tightened up so several descriptions of the reviewed publications are synthesized? As it stands now, much of the review reads like individual summary sentences of many studies have been placed together, rather than having a clear narrative and flow.

Generally, the english and sentence structures are strong and the review is an important contribution to the field. I look forward to reading a tighter and more integrated version!

Author Response

This review discusses the association of MS with other autoimmune disease co-morbidities. Given the limited and conflicting literature in this field it is a challenging undertaking to write a cohesive review on this topic. That being said, there could be substantial improvement in the organization and structure of the review. There is a lot of repetition that could be minimized to make the review much shorter and concise (making it an "easier read").

There are many stand alone thoughts or paragraphs. Could this be tightened up so several descriptions of the reviewed publications are synthesized? As it stands now, much of the review reads like individual summary sentences of many studies have been placed together, rather than having a clear narrative and flow.

Generally, the english and sentence structures are strong and the review is an important contribution to the field. I look forward to reading a tighter and more integrated version!

We thank the Referee for her/his important suggestion. Accordingly, we have made changes to the manuscript to make it clearer, and we added headings to the paragraphs to avoid stand-alone sentences. We also added a separate paragraph focusing on multiple sclerosis and neurological comorbidities and added a table to synthesize the studies.

Round 2

Reviewer 3 Report

Edits have improved the manuscript.